# Rigorous bounds on dynamical response functions and time-translation symmetry breaking

**Marko Medenjak[1⋆], Tomaz Prosen[2] and Lenart Zadnik[3]**

**1** Institut de Physique Théorique Philippe Meyer, École Normale Supérieure, PSL University, Sorbonne Universités, CNRS, 75005 Paris, France
**2** Faculty of Mathematics and Physics, University of Ljubljana, Jadranska 19, SI-1000 Ljubljana, Slovenia
**3** Université Paris-Saclay, CNRS, LPTMS, 91405, Orsay, France

⋆ medenjak@lpt.ens.fr

## Abstract

Dynamical response functions are standard tools for probing local physics near the equilibrium. They provide information about relaxation properties after the equilibrium state is weakly perturbed. In this paper we focus on systems which break the assumption of thermalization by exhibiting persistent temporal oscillations. We provide rigorous bounds on the Fourier components of dynamical response functions in terms of extensive or local dynamical symmetries, i.e., extensive or local operators with periodic time dependence. Additionally, we discuss the effects of spatially inhomogeneous dynamical symmetries. The bounds are explicitly implemented on the example of an interacting Floquet system, specifically in the integrable Trotterization of the Heisenberg XXZ model.

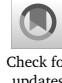

# 1 Introduction

Response functions, or susceptibilities, can be used to probe the symmetry breaking phenomena. In analogy with static susceptibilities, which probe transition from the ordered to disordered phase and, for example, characterize the spontaneous breaking of the space-translation symmetry in crystals [1], the dynamical susceptibilities carry information about the dynamical phases of matter. For instance, ideal conductivity, which is a particular manifestation of ergodicity breaking, can be related to the Drude weight, which corresponds to the zero-frequency behavior of the dynamical response function [2]. More generally, non-vanishing dynamical response functions at finite frequencies imply breaking of time-translation symmetry of equilibrium states.

Obtaining rigorous or explicit results in quantum strongly interacting many-body systems is a formidable task even in the presence of "exact" solvability. Typically one has to rely on numerical simulations, which, however, are bound to fail, either due to finite-size effects or because of the entanglement growth. One of the most important rigorous results explaining the origins of non-ergodic behavior is Mazur's lower bound on asymptotics or time-averages of dynamical correlation functions [3]. Through the scope of this bound, non-ergodicity can be understood as a consequence of underlying *extensive* or *local symmetries* of the system. In particular, local symmetries can be related to localization phenomena in many-body systems [4–6], while extensive symmetries lead to ideal transport at arbitrary temperature and lack of thermalization in integrable models [2,7–9].

A large amount of recent publications deal with systems that avoid relaxation to equilibrium. Roughly speaking, they can be divided into two categories. The first class comprises quantum scarred models [10–13] that avoid relaxation for special but physically relevant initial conditions. The second class of systems that defy relaxation to equilibrium are quantum time crystals [14–32], where time-translation symmetry breaking occurs for typical states, or equivalently, on the level of dynamical response functions [30,33].

The emergence of time-translation symmetry breaking in strongly interacting systems has recently been related to extensive dynamical symmetries [30], calling for the development of a rigorous framework. In this paper we provide such a framework by deriving strict lower bounds on AC dynamical response functions in terms of dynamical symmetries. The lower bounds are derived for autonomous Hamiltonian dynamics as well as for periodically driven (Floquet) systems. We apply our results to a nontrivial example of a many-body Floquet system that breaks the discrete time-translation symmetry, specifically to the integrable Trotterization of the spin-1/2 XXZ model [34,35].

# 2 Breaking of time-translation symmetry

Breaking of the time-translation symmetry is associated with a failure of a perturbed stationary state to return to stationarity, even in the infinite-time limit $t \to \infty$. Analogously, we can consider the space-translation symmetry breaking from a dynamical point-of-view: it occurs, when the information about spatial inhomogeneities induced by the perturbation is retained indefinitely after the perturbation has been switched off. In general we will consider stationary states $\rho(\underline{\mu})$, where $\underline{\mu}$ denotes the set of chemical potentials. Typically, only few chemical potentials describe the complete steady-state manifold pertaining to the system. In Hamiltonian systems, one of the chemical potentials is the inverse temperature $\mu_1 = \beta$. Assuming that the (unnormalized) thermal state of the system $\rho(\beta) = \exp(-\beta H)$ is slightly perturbed at $t = 0$, say by a Hermitian operator $B$, i.e., $H \to H - (\delta/\beta)B$, we probe the dynamics of the local

observable $A$ by considering the first order

$$\partial_\delta \langle A(t) \rangle_\delta \big|_{\delta=0} = \int_0^\beta \frac{\mathrm{d}\lambda}{\beta} \langle A(t)\rho(\beta-\lambda)B\rho(\beta-\lambda)^{-1} \rangle - \langle A \rangle \langle B \rangle \tag{1}$$

in the expansion of its expectation value. Here, we have introduced the averages with respect to the perturbed, $\langle A \rangle_\delta = \mathrm{tr}(A\rho)/\mathrm{tr}(\rho)$, and thermal state, $\langle A \rangle \equiv \langle A \rangle_0$. The response of expectation values to a perturbation can be interpreted in terms of the canonical Kubo-Mori-Bogoliubov inner product [36]

$$\langle A, B \rangle = \int_0^\beta \frac{\mathrm{d}\lambda}{\beta} \langle A^\dagger \rho(\beta-\lambda)B\rho(\beta-\lambda)^{-1} \rangle - \overline{\langle A \rangle} \langle B \rangle. \tag{2}$$

Physically sensible perturbations of extensive Hamiltonians $H$ are either extensive, e.g., a sum of local operators acting on a spin lattice, or themselves local.

In order to probe the frequency dependence of the response, it is useful to introduce the dynamical response function

$$f_{AB}(\omega) = \lim_{T \to \infty} \frac{1}{2T} \int_{-T}^T \mathrm{d}t\, e^{\mathrm{i}\omega t} \langle A(t), B \rangle. \tag{3}$$

At zero frequency it represents the time-averaged perturbative correction (1) to $\langle A \rangle_0$. Now, the system is called non-thermal, provided that $f_{AB}(0) \neq 0$ and $f_{HB}(0) = 0$. The condition $f_{HB}(0) = 0$ ensures that the energy is conserved in the first order of the perturbative expansion, and in turn implies that the thermal ensemble with associated expectation values should remain the same under the assumption of thermalization. However, due to $f_{AB}(0) \neq 0$, the time average of the expectation value does not coincide with the old thermal average $\langle A \rangle_0$. If, in addition, the finite-frequency response function does not vanish, i.e., $f_{AB}(\omega) \neq 0$, for some $\omega \neq 0$, the system does not relax to *any* stationary distribution, for arbitrarily weak perturbation of the equilibrium ensemble. This, then, characterizes the breaking of time-translation symmetry.

Similarly, the breaking of space-translation symmetry on the lattice can be probed by spatially modulated extensive observables

$$A_k = \sum_n e^{\mathrm{i}kn} a_n, \tag{4}$$

with local densities $a_n$, via the frequency and wavevector-dependent response function

$$f_{AB}(k, \omega) = \lim_{T \to \infty} \frac{1}{2T} \int_{-T}^T \mathrm{d}t\, e^{\mathrm{i}\omega t} \langle A_k(t), B_k \rangle. \tag{5}$$

As before, the nonvanishing susceptibility $f_{AB}(k, \omega) \neq 0$ for a nonzero value of the wavevector, $k \neq 0$, implies that the system will retain memory related to the spatial modulation of the perturbation. Here, we have assumed that the Hamiltonian $H$ is translationally invariant, implying $\langle A_k(t), B_{k'} \rangle = 0$, if $k \neq k'$. If the stationary state $\langle \bullet \rangle$ is not translationally invariant, one might wish to study also the off-diagonal elements $\lim_{T \to \infty} \frac{1}{2T} \int_{-T}^T \mathrm{d}t\, e^{\mathrm{i}\omega t} \langle A_k(t), B_{k'} \rangle$ for different $k$ and $k'$.

Note that our definitions (3) and (5) differ from the standard definitions of dynamical response functions by an extra factor of $1/T$ in the Fourier transformation. This factor ensures that $f_{AB}(\omega)$ is finite, and not a Dirac delta singularity, for a perfectly harmonic response at frequency $\omega$.

# 3 Local and extensive dynamical symmetries

Analogously to how the non-thermal behavior can be understood as a consequence of local and extensive conservation laws [8], the time-translation symmetry breaking relates to the existence of local and extensive dynamical symmetries [30].

For a concise presentation let us revisit the notions of locality and extensivity, as they play a prominent role in our story. We consider extended quantum systems defined on a regular lattice of $N$ sites with finite-dimensional local Hilbert space. Prominent examples of such systems are spin chains or spin lattices. The notions of (effective) locality and extensivity are, in general, state-dependent [37]. Operator $a$ is local with respect to the thermal state $\rho$ (or generic clustering state that is invariant under the time evolution), if its Kubo-Mori-Bogoliubov norm is finite in the thermodynamic limit

$$0 < \lim_{N \to \infty} \langle a, a \rangle < \infty. \tag{6}$$

Operator $A$ is extensive with respect to the state $\rho$, if its norm is proportional to the volume of the system (i.e., the number of local physical sites, $N$) in the thermodynamic limit,

$$0 < \lim_{N \to \infty} \frac{1}{N} \langle A, A \rangle < \infty, \tag{7}$$

and has a nonvanishing overlap with at least one local operator $b$

$$\lim_{N \to \infty} \langle b, A \rangle > 0. \tag{8}$$

Extensivity, as defined here, is sometimes also referred to as *pseudo-locality* [38]. While extensive dynamical symmetries are of central importance when considering extensive perturbations of stationary states, local dynamical symmetries are crucial when dealing with local perturbations. Such symmetries should be employed to study discrete time crystals that can arise in many-body localized systems [15]. In systems with multiple extensive conserved quantities $Q_j$, i.e., $[H, Q_j] = 0$, the array of equilibrium ensembles is naturally enlarged and local observables are expected to relax to their equilibrium values, described by the associated set of generalized inverse temperatures (or chemical potentials) $\beta_j$ [39, 40]. The set of stationary states can be rigorously established using extensive charges as flows on the space of operators [37]. Such systems behave non-thermally, and are well-described by non-thermal maximum-entropy states, the so-called generalized Gibbs states $\rho_{GGE} = \exp(-\sum_j \beta_j Q_j)$, provided that the complete set of extensive integrals of motion $Q_j$ has been identified. Upon a slight perturbation of the state $\rho_{GGE}$ by an extensive operator, the expectation values of local observables will relax back to a slightly perturbed stationary state $\rho'_{GGE}$. Similarly, if the system admits local (non-extensive) integrals of motion $q_j$, $[H, q_j] = 0$, it fails to relax to the thermal state even if the perturbation of the initial stationary ensemble is local.

In order to be considered as extensive (or local) *dynamical* symmetries, operators $Q_j$ should satisfy the definition of extensivity (or locality), as well as the eigenoperator condition

$$[H, Q_j] = \omega_j Q_j. \tag{9}$$

Clearly, extensive dynamical symmetries admit a simple periodic time-dependence $Q_j(t) = \exp(i \omega_j t) Q_j$. The simplest example are dynamical symmetries responsible for the spin precession occurring in magnets with SU(2) invariant interaction $H_{\text{SU}(2)}$ in the presence of external magnetic field $S^z = \sum_n s_n^z$, i.e., $H = H_{\text{SU}(2)} + h S^z$. They are given by the remaining generators of the $sl_2$ algebra, $S^\pm = \sum_n s_n^\pm$, which satisfy $[H, S^\pm] = \pm h S^\pm$. From a more general perspective, systems with dynamical symmetries can be obtained from models described by a

Hamiltonian $H$ with at least a pair of non-abelian local symmetries, $[H,X] = 0$, $[H,Y] = 0$, that form a closed algebra, for instance $[X,Y] = \alpha Y$. Then, $Y$ is the dynamical symmetry for the system described by $H' = H + \gamma X$, with the corresponding frequency $\omega = \alpha\gamma$. The existence of systems with dynamical symmetries that cannot be obtained in this manner is unknown.

The notion of (dynamical) symmetries can be trivially extended to non-autonomous time-periodic (Floquet) systems. In this case the dynamics of local observable $a$ is generated by a periodic, time-dependent Hamiltonian $H(t) = H(t + \tau)$

$$\partial_t a = \mathrm{i}[H(t), a], \tag{10}$$

with a period $\tau$. In the case of time dependent Hamiltonians we will focus on the stroboscopic time evolution of observables

$$\hat{\mathcal{M}}_\tau[a] = \overrightarrow{\exp}\left(\mathrm{i}\int_0^\tau \mathrm{d}t\, H(t)\right) a \overleftarrow{\exp}\left(-\mathrm{i}\int_0^\tau \mathrm{d}t\, H(t)\right), \tag{11}$$

where $\overleftarrow{\exp}$ is the time-ordered exponential, the arrow denoting the direction of increasing time in the time-ordering. In the case of non-autonomous systems, local or extensive dynamical symmetries can again be defined by locality or extensivity and quasi-periodicity of their time-dependence

$$\hat{\mathcal{M}}_\tau[Q_j] = \exp(\mathrm{i}\omega_j\tau)Q_j. \tag{12}$$

Quasi-periodicity arises since $\omega_j\tau/(2\pi)$ is, in general, not a rational number, meaning that, for any integer $n > 0$, $(\hat{\mathcal{M}}_\tau)^n[Q_j] \neq Q_j$.

Thermalization and equilibration in quantum systems can be understood through the scope of *eigenstate thermalization hypothesis* (ETH). It is thus instructive to clarify the role of dynamical symmetries also in this regard. Clustering property of the initial state ensures that the dynamics is restricted to states within a narrow window of energy and expectation values of other extensive conservation laws. After a short time, the off-diagonal elements of the time-evolved observables vanish due to dephasing and suppression in the thermodynamic limit, and we observe the onset of stationarity, which can be described either by a representative eigenstate, or equivalently, by a statistical ensemble. Systems with dynamical symmetries avoid equilibration by circumventing two assumptions of the ETH. As a consequence of the eigen-operator condition (9), there exists an eigenstate $|\psi'\rangle$ corresponding to the energy $E + \omega_j$, for any eigenstate $|\psi\rangle$ with energy $E$, provided that $Q_j|\psi\rangle \neq 0$ (then, in fact, $Q_j|\psi\rangle \propto |\psi'\rangle$). This implies that (i) dephasing does not occur for eigenstates with the energy difference of the order $\mathcal{O}(1)$ and (ii) that the corresponding off-diagonal elements are not exponentially suppressed in the thermodynamic limit, resulting in perpetual oscillations of some local (or extensive) observables.

## 4 Bounds on susceptibilities

The purpose of this section is twofold. We first generalize the bound provided by Mazur [3] to AC response functions and secondly, provide general bounds on the off-diagonal components of the susceptibility matrix. We will first focus on time-independent Hamiltonian systems. Let us start by considering the following operator

$$O_A(\omega) = \frac{1}{T}\int_0^T \mathrm{d}t\, e^{-\mathrm{i}\omega t}A(t) - \sum_j \alpha_j Q_j, \tag{13}$$

where $Q_j$ are either extensive or local dynamical symmetries. To obtain the lower bound on the Fourier components we consider the Kubo-Mori-Bogoliubov norm (1) of the operator (13), with respect to some stationary state $\rho(\beta)$, which is clearly non-negative

$$\langle O_A(\omega), O_A(\omega)\rangle \geq 0. \tag{14}$$

Writing out all of the components explicitly, we obtain the bound

$$\frac{1}{T^2}\int_0^T dt_1 \int_0^T dt_2\, e^{i\omega(t_1-t_2)}\langle A(t_1),A(t_2)\rangle \geq$$
$$\geq \sum_j \frac{1}{T}\int_0^T dt\left(e^{i(\omega-\omega_j)t}\alpha_j\langle A,Q_j\rangle + c.c.\right) - \sum_{j,l}\bar{\alpha}_j\alpha_l\langle Q_j,Q_l\rangle. \tag{15}$$

To study the susceptibilities in the thermodynamic limit $N \to \infty$, the latter has to be taken after dividing each term by the system size $N$. Equivalently, we can, in this case, substitute the Kubo-Mori-Bogoliubov bracket $\langle A,B\rangle$ by $\lim_{N\to\infty}\langle A,B\rangle/N$. Only then we take the infinite-time limit $T \to \infty$ of each term in the inequality (15).

The optimal set of coefficients $\alpha_j$ can be obtained by maximizing the quadratic form on the right-hand side of Eq. (15), yielding the set of equations

$$\delta_{\omega,\omega_j}\langle Q_j,A\rangle = \sum_l \langle Q_j,Q_l\rangle\,\alpha_l, \tag{16}$$

where the Kronecker delta on the left-hand side emerges from time-averaging

$$\delta_{\omega,\omega_j} = \lim_{T\to\infty}\frac{1}{T}\int_0^T dt\,\exp(-i[\omega-\omega_j]t). \tag{17}$$

Remember that in the standard definition of response functions the prefactor $1/T$ is absent, whence the integral over time instead results in the Dirac delta peak. We remark that the overlaps $\langle Q_j,Q_l\rangle$ in Eq. (16) are nonzero only if $\omega_j = \omega_l$, which follows from the time-translation invariance of the thermal state. It thus suffices to consider only the dynamical symmetries $Q_j$ that correspond to the same frequency $\omega_j \equiv \omega_l$ in the eigenoperator condition (9).

In order to represent the results compactly, we now introduce a hermitian matrix of kernels $\mathcal{Q} = \mathcal{Q}^\dagger$, with elements $\mathcal{Q}_{j,l} = \langle Q_j,Q_l\rangle$. Additionally, we require the knowledge of overlaps between the observable $A$ and dynamical symmetries $Q_j$, namely $\mathcal{A}_j = \delta_{\omega,\omega_j}\langle Q_j,A\rangle$, gathered into the vector $\mathcal{A}$. Plugging the solution of the set of equations (16) into Eq. (15) and taking the limit $T \to \infty$, yields the lower bound

$$\lim_{T\to\infty}\frac{1}{T^2}\int_0^T dt_1 \int_0^T dt_2\, e^{i\omega(t_1-t_2)}\langle A(t_1),A(t_2)\rangle \geq \mathcal{A}^\dagger\mathcal{Q}^{-1}\mathcal{A}. \tag{18}$$

The left-hand side of the above equation can be further simplified to

$$\lim_{T\to\infty}\frac{1}{2T}\int_{-T}^T dt\, e^{i\omega t}\langle A(t),A\rangle, \tag{19}$$

if a weak requirement that the Fourier components of the dynamical response function asymptotically approach their averaged value

$$\lim_{T\to\infty}\frac{1}{T}\int_{-T}^T dt\,\frac{|t|}{T}e^{i\omega t}\langle A(t),A\rangle = \lim_{T\to\infty}\frac{1}{2T}\int_{-T}^T dt\, e^{i\omega t}\langle A(t),A\rangle, \tag{20}$$

is satisfied. The lower bound thus reads

$$f_{AA}(\omega) \geq \mathcal{A}^\dagger \mathcal{Q}^{-1} \mathcal{A}. \tag{21}$$

The result (21) straightforwardly generalizes to the off-diagonal elements of the susceptibility tensor. Specifically, considering a pair of different observables $A, B$ and defining the overlaps $\mathcal{B}_j = \delta_{\omega,\omega_j} \langle Q_j, B \rangle$, we obtain the following bound

$$e^{i\phi} f_{AB}(\omega) \geq e^{i\phi} \mathcal{A}^\dagger \mathcal{Q}^{-1} \mathcal{B} \geq 0, \tag{22}$$

where the phase $\phi$ is chosen appropriately, to render the right-hand-side of the first inequality (22) real and nonnegative. In order to arrive at the bound (22), we substituted $A \to \alpha A + \beta e^{i\phi} B$ in the inequality (21) and took the derivatives w.r.t. $\alpha$ and $\bar{\beta}$ at $\alpha = \beta = 0$. Note that the above result can be further generalized to systems with spatially modulated response functions

$$e^{i\phi} f_{AB}(k, \omega) \geq e^{i\phi} \mathcal{A}_k^\dagger \mathcal{Q}_k^{-1} \mathcal{B}_k, \tag{23}$$

where the overlaps $\mathcal{A}_k$, $\mathcal{Q}_k$, and $\mathcal{B}_k$ contain only the contributions of spatially modulated extensive symmetries $Q_j$, associated with the wave vector k, i.e.,

$$\hat{\mathcal{S}}[Q_j] = e^{ik} Q_j. \tag{24}$$

Here, $\hat{\mathcal{S}}$ denotes conjugation by a single-site lattice shift (one-site translation).

In the discrete-time (Floquet) case the dynamical susceptibility can be defined as

$$f_{AB}(k, \omega) = \lim_{L \to \infty} \frac{1}{2L} \sum_{l=-L}^{L} e^{i\omega l \tau} \langle A_k(l\tau), B_k \rangle, \tag{25}$$

where $l \in \mathbb{Z}$ counts the steps in the stroboscopic time evolution. In this case the lower bounds can be obtained by replacing the continuous-time variable with stroboscopic steps, $t \to l\tau$, and time averages with sums, $\lim_{T \to \infty} \frac{1}{2T} \int_{-T}^{T} dt \to \lim_{L \to \infty} \frac{1}{2L} \sum_{l=-L}^{L}$, in the above derivations.

# 5 Driven Heisenberg chain at root-of-unity anisotropies

To illustrate how persistent oscillations of extensive many-body observables occur, we consider a Floquet driven spin-1/2 chain of even size $N \in 2\mathbb{N}$, introduced in Refs. [34, 35] as an integrable Trotterization of the anisotropic Heisenberg (XXZ) model. For simplicity we assume that the thermal ensemble corresponds to the infinite-temperature state $\rho(0) = 2^{-N} \mathbb{1}$, although the results can be generalized to arbitrary time-translation invariant state generated by extensive conserved quantities of the model. We fix a unit period $\tau = 1$ here, and consider a discrete (stroboscopic) evolution, where time $t$ is an integer. Specifically, for an arbitrary observable $A$ we define dynamical map $A(t+1) = \hat{\mathcal{M}}_1[A(t)]$ where

$$\hat{\mathcal{M}}_1[A(t)] = U^{-1} A(t) U. \tag{26}$$

The two half-steps of the propagator $U = U_o U_e$ read

$$U_e = U_{1,2} \dots U_{N-1,N}, \qquad U_o = U_{2,3} \dots U_{N,1}. \tag{27}$$

Each of them comprises local unitary quantum gates

$$U_{n,n+1} = e^{-i\left[J(s_n^x s_{n+1}^x + s_n^y s_{n+1}^y) + \Delta(s_n^z s_{n+1}^z - \mathbb{1}) + \frac{h}{2}(s_n^z + s_{n+1}^z)\right]}, \tag{28}$$

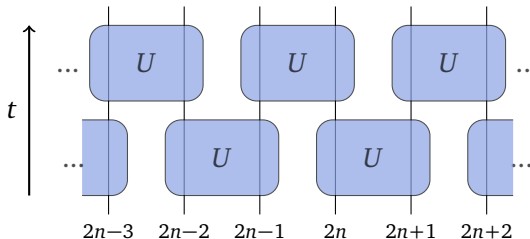

Figure 1: One time step of the discrete time evolution of an observable, represented by the propagator $U = U_o U_e$. The half-steps are given in (27) and the time flows upwards.

acting on pairs of neighboring sites. Operators $s^\alpha$, for $\alpha \in \{x, y, z\}$, denote the spin-1/2 matrices. The circuit representation of a single time step is shown in Fig. 1

Here we focus on an interesting regime of the model characterized by (i) extensive symmetries that break the spin-flip ($\mathbb{Z}_2$) symmetry, and more importantly, (ii) extensive dynamical symmetries that break its U(1) invariance, corresponding to the conservation of magnetization. While the first set of symmetries prevents decay of spin-current fluctuations and leads to ideal spin transport [35], the second set, which is the focus of this paper, prevents relaxation of local observables that couple different magnetization sectors, provided that the magnetic field is nonzero, i.e., $h \neq 0$. Since dynamical symmetries are related to the integrability structure of the model, it proves useful to parameterize the Floquet operator in terms of real *anisotropy parameter* $\gamma \in \mathbb{R}$ and imaginary *staggering* $\delta \in i\mathbb{R}$

$$\Delta = \frac{1}{i} \log\left[\frac{\sin(\gamma - \delta)}{\sin(\gamma + \delta)}\right], \qquad J = \frac{1}{i} \log\left[\frac{\sin\gamma - \sin\delta}{\sin\gamma + \sin\delta}\right]. \tag{29}$$

Through this mapping the local unitary gate (28) is related to the trigonometric *R*-matrix, which constitutes the local conservation laws of the XXZ model; see Appendix A for the review of the integrability structure. In order to reproduce the continuous-time Heisenberg evolution, the propagator (26) should be expanded in $\delta$ to the leading order.

Dynamical symmetries $Y(\lambda)$ of the model are continuously parametrized: the discrete index $j$ in Eq. (12) is substituted by a complex *spectral parameter* $\lambda \in \mathbb{C}$. They stem from the so-called *semicyclic* complex-spin representation of the XXZ model's symmetry group that lacks the lowest-weight state and exists only for

$$\gamma \in \left\{\pi \frac{\ell}{m} \,\middle|\, m + 1, \ell \in 2\mathbb{N}, \ \ell < m\right\}, \tag{30}$$

that is, for commensurable anisotropies of odd order $m$ [30,41,42]. In the absence of magnetic fields, i.e., for $h = 0$, these operators are invariant under the time evolution and form a family of extensive conservation laws of the Floquet driven XXZ model; the reader is invited to consult Appendix B for elaborate details on their construction. In short, the semicyclic representation of the symmetry group is spanned by $m \times m$ matrices $\mathbf{S}^\alpha(\beta)$, $\alpha \in \{0, +, -, z\}$, which depend on the real representation parameter $\beta$ and encode coefficients in the operator-basis expansion of $Y(\lambda)$ according to[1]

$$Y(\lambda) = \sum_{\alpha_1, \dots, \alpha_N} \partial_\beta \langle 0 | \mathbf{S}^{\alpha_1}(\beta) \dots \mathbf{S}^{\alpha_N}(\beta) | 0 \rangle \big|_{\beta=0} s_1^{\alpha_1} \dots s_N^{\alpha_N}. \tag{31}$$

---

[1]Eq. (31) serves only for illustrative purposes, as many details have been disregarded. The accurate matrix-product form of $Y(\lambda)$ is presented in Appendix B.

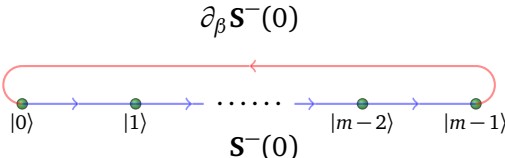

Figure 2: Combined periodic action of $\mathbf{S}^-(0)$ (in blue) and $\partial_\beta \mathbf{S}^-(0)$ (in red) on the auxiliary degree of freedom. Operator $\mathbf{S}^-(\beta)$ is one of the matrices that encode the coefficients in the operator-basis expansion (31) of the dynamical symmetries $Y(\lambda)$. Since it is coupled to the spin raising matrix $s^+$, its $m$-step periodic action connecting the state $|0\rangle$ to itself produces a surplus of $m$ operators $s^+$ acting on different sites in the spin chain.

Here, $s^0 = \mathbb{1}$ denotes a $2 \times 2$ identity matrix and $s^\pm = s^x \pm is^y$, while $|0\rangle$ is the highest-weight state of the semicyclic representation, used to project out the auxiliary degree of freedom, upon which operators $\mathbf{S}^\alpha(\beta)$ act. For $\beta = 0$ these operators become tridiagonal, however, due to the absence of the lowest-weight state in the auxiliary space, $\mathbf{S}^-(0)$ and $\partial_\beta \mathbf{S}^-(0)$ together act periodically, as sketched in Fig. 2. This periodicity lies at the origin of the U(1) symmetry breaking.[2]

Due to this periodic action, each term of the continuously-parametrized extensive symmetries $Y(\lambda)$ possesses a surplus of $m$ spin raising operators $s^+$. In the presence of magnetic fields this results in the oscillatory time-dependence

$$\mathrm{U}^{-t} Y(\lambda) \mathrm{U}^t = e^{ihmt} Y(\lambda) \tag{32}$$

with a frequency, proportional to the order of the anisotropy $m$ and the magnetic field strength $h$. Note that the evolution equation (32) associates $Y(\lambda)^\dagger$ to the negative frequency $\omega = -hm$. Since all terms in $Y(\lambda)^\dagger$ contain a surplus of $m$ spin-lowering operators $s^-$, the continuous family $\{Y(\lambda)^\dagger\}$ of adjoint dynamical symmetries is independent of $\{Y(\lambda)\}$ and should be included in the bounds on the Fourier components of dynamical susceptibilities.

In Appendix B we show that $Y(\lambda)$ are extensive in the size of the system, whenever $|\mathrm{Re}\lambda - \pi/2| < \pi/(2m)$. In this strip in the complex plane one can compute their overlap $\mathcal{Y}(\lambda,\mu) = \langle Y(\bar{\lambda}), Y(\mu)\rangle$ according to a conjectured and numerically thoroughly checked formula[3]

$$\mathcal{Y}(\lambda,\mu) = \frac{\big(\cos(\lambda - \mu + \delta) + \cos(\lambda - \mu - \delta) - 2\cos(\lambda + \mu)\big)\sin(\lambda + \mu)}{2(\sin\gamma)^2(\cos 2\lambda - \cos\delta)(\cos 2\mu - \cos\delta)\sin(m[\lambda + \mu])}, \tag{33}$$

which is essential in the computation of the bounds on the dynamical susceptibilities. We remind the reader that the inner product has been rescaled by the system size $N$, and the thermodynamic limit $N \to \infty$ has been taken.

To relate to the discussion of the dynamical susceptibilities, we will now consider a set of extensive observables

$$A = 2^m \sum_{n=1}^{N} \prod_{r=0}^{m-1} s_{n+r}^x, \tag{34}$$

---

[2]The term *highest-weight state* comes from the fact that $|0\rangle$ is annihilated by the spin raising operator $\mathbf{S}^+(\beta)$ on the auxiliary space (see Appendix B).

[3]The formula is conjectured similarly as in Ref. [35], by considering various limits with less elaborate calculation of the overlap $\langle Y(\bar{\lambda}), Y(\mu)\rangle$, such as the continuous-time limit [41].

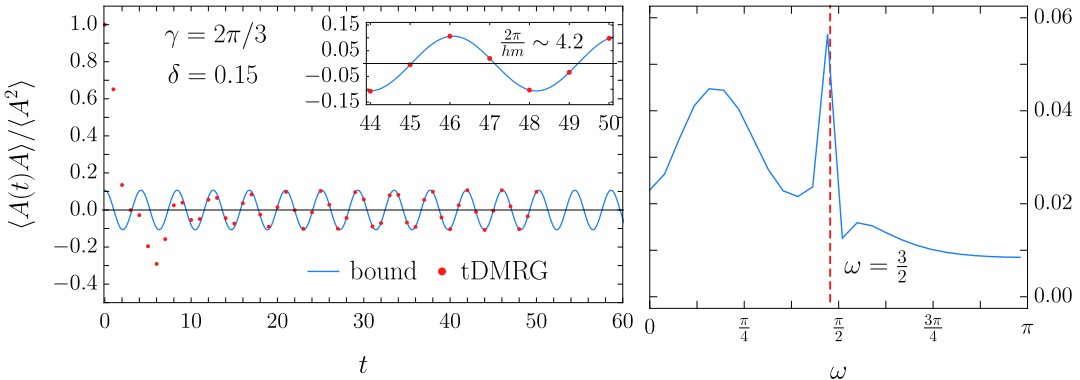

Figure 3: In the left figure we plot numerical tDMRG data (red dots) and analytical result (blue line) for the Fourier component of the dynamical susceptibility. Parameters are $\gamma = 2\pi/3$ and $\delta = 0.15$. Strength of the magnetic field is $h = 1/2$ and the frequency of oscillations is $hm = 3/2$. The period of oscillation is clearly discernible in the inset. In the figure on the right, we plot the power spectrum obtained from the tDMRG data on the left, and denote the expected frequency of oscillations by a red dotted line.

overlapping with dynamical symmetries $Y(\lambda)$ and their adjoint counterparts $Y(\lambda)^\dagger$. The linear-response susceptibility of observable $A$ evidently possesses nonzero Fourier components associated with frequencies $\omega = \pm hm$, arising from terms $\prod_{r=0}^{m-1} s_{n+r}^{\pm}$. They can be bounded from below by means of the system of integral equations

$$\int d\mu\, \mathcal{Y}(\lambda,\mu)y(\mu) = \mathcal{A}(\lambda), \qquad f_{AA}(\omega) \geq D = \int d\lambda\, y(\lambda)\overline{\mathcal{A}(\bar{\lambda})}, \tag{35}$$

which generalize the bound (21) to the case, where dynamical symmetries are enumerated by a continuous parameter $\lambda$ instead of a discrete index (following Ref. [38]). The projection $\mathcal{A}(\lambda) = \langle Y(\bar{\lambda}), A\rangle$ onto the dynamical symmetries reads

$$\mathcal{A}(\lambda) = \frac{\csc(\lambda_+) + \csc(\lambda_-)}{4^{\frac{m+1}{2}}[\sin(\lambda_-)\sin(\lambda_+)]^{\frac{m-1}{2}}} \prod_{k=2}^{m-1}\sin(k\gamma) \tag{36}$$

and can be discerned from the matrix product form of $Y(\lambda)$, shown in Appendix B.

Assuming that $\{Y(\lambda)\}$ and $\{Y(\lambda)^\dagger\}$ form a complete set of dynamical symmetries, the lower bound, computed by solving the linear integral equation (35), saturates. We can then conjecture the asymptotic behaviour of the temporal auto-correlation function to be

$$\frac{\langle A(t)A\rangle}{\langle A^2\rangle} \sim 2D\cos(hmt). \tag{37}$$

This result is indeed corroborated by the numerical evidence obtained via the time-dependent density-matrix renormalization group method (tDMRG) shown in Fig. 3.

The time translation symmetry breaking is not specific to the Floquet driven XXZ model, but can be observed also in its continuous-time limit; see Ref. [30] for an elaborate discussion. The corresponding dynamical symmetries are simply $Y(\lambda)$ evaluated at $\delta = 0$, while the dynamical equation then reads

$$[H, Y(\lambda)] = hmY(\lambda). \tag{38}$$

# 6 Conclusion

In this article we derived rigorous bounds on AC dynamical response functions, providing insight into the spontaneous time-translation symmetry breaking of the underlying dynamics. The research outlines a rigorous approach towards the study of quantum time crystals, and probing of the spatial symmetry breaking phenomena, such as dimerization, by shedding the light on its microscopic origins.

From the broader point of view our study centers on types of information that can be preserved in interacting many-body quantum systems, and might thus prove lucrative from the point of view of quantum information processing and storage. From this perspective, finding concrete examples of systems, which go beyond the toy models presented in this paper, is of paramount importance. There are two somewhat related properties that should be at the forefront of this endeavour. The first one is stability to perturbations, and the second one the viability of implementing these systems in experimental setups.

While we focused on close-to-equilibrium setup in this paper, the dynamical symmetries have a profound effect on the long-time description when the system is initially prepared in the state that is far from equilibrium as well. It was conjectured that, in this case, the appropriate description is provided by time dependent maximum entropy ensembles [30], however any clear theoretical account of this phenomena is yet to be provided.

It was recently proposed [43–45] that less-local conservation laws might have effects on the equilibration rate and normal conductivity. The effects of, for example, quadratically extensive dynamical symmetries is yet to be explored in this context.

# Acknowledgements

M. M. thanks B. Buča for numerous discussions on related topics. DMRG calculations were performed using the ITensor Library [46].

**Funding information** T. P. acknowledges support by Program P1-0402 of Slovenian Research Agency (ARRS), and Advanced grant OMNES No. 694544 of European Research Council (ERC). L. Z. acknowledges also the support by the European Research Council under the Starting Grant No. 805252 LoCoMacro.

# Appendix

## A Integrability of the driven XXZ model

In this appendix we comment on the integrability structure of the Floquet driven XXZ spin-1/2 chain. Under the parameter mapping (29), the local quantum unitary gate (28) transforms into the trigonometric $R$-matrix of the spin-1/2 XXZ model [35]. Explicitly

$$U_{n,n+1} = P_{n,n+1}R_{n,n+1}(\delta)e^{-i\frac{h}{2}(s_n^z+s_{n+1}^z)}, \tag{39}$$

where $P_{n,n+1}$ is a permutation (transposition) that exchanges the one-particle states of two neighbouring spins, while the trigonometric $R$-matrix reads

$$R(\lambda) = \begin{bmatrix} 1 & 0 & 0 & 0 \\ 0 & \frac{\sin\lambda}{\sin(\lambda+\gamma)} & \frac{\sin\gamma}{\sin(\lambda+\gamma)} & 0 \\ 0 & \frac{\sin\gamma}{\sin(\lambda+\gamma)} & \frac{\sin\lambda}{\sin(\lambda+\gamma)} & 0 \\ 0 & 0 & 0 & 1 \end{bmatrix}. \tag{40}$$

For $h = 0$, the propagator $U = U_o U_e$ with half-steps (27) can be recovered from the integrability structure as

$$U = [T(-\tfrac{\delta}{2})]^{-1} T(\tfrac{\delta}{2}), \tag{41}$$

where we have introduced a continuous family of transfer matrices

$$T(\lambda) = \mathrm{tr}_0[R_{0,1}(\lambda_+)R_{0,2}(\lambda_-)\ldots R_{0,N}(\lambda_-)], \tag{42}$$

in which $\lambda_\pm = \lambda \pm \delta/2$ denote shifts in the spectral parameter. As a consequence of the Yang-Baxter equation

$$R_{1,2}(\lambda-\mu)R_{1,3}(\lambda)R_{2,3}(\mu) = R_{2,3}(\mu)R_{1,3}(\lambda)R_{1,2}(\lambda-\mu), \tag{43}$$

satisfied by the trigonometric $R$-matrix (40), the transfer operators commute for all pairs of spectral parameters $\lambda$ and $\mu$, i.e., $[T(\lambda), T(\mu)] = 0$. This establishes integrability of the model, which holds also when $h \neq 0$, due to the $U(1)$ symmetry

$$[e^{i\frac{h}{2}(s_n^z + s_{n+1}^z)}, R_{n,n+1}(\delta)] = 0 \tag{44}$$

of the trigonometric $R$-matrix (40). The standard hierarchy of local conservation laws is produced by logarithmic derivatives of the transfer matrix (42), namely

$$Q_j^\pm = \partial_\lambda^j \log T(\lambda)\,|_{\lambda=\pm\frac{\delta}{2}}\,. \tag{45}$$

## B  Extensive dynamical symmetries in the driven XXZ model

The extensive dynamical symmetries in the driven XXZ spin-1/2 model originate in the transfer matrix

$$\tilde{T}(\lambda) = \mathrm{tr}_a\big[\mathbf{L}_{a,1}(\lambda_+)\mathbf{L}_{a,2}(\lambda_-)\ldots\mathbf{L}_{a,N}(\lambda_-)\big], \tag{46}$$

in which $R$-matrices have been substituted by Lax operators

$$\mathbf{L}_{a,n}(\lambda) = \mathbb{1}\cos[\gamma\mathbf{S}_a^z] + 2\cot\lambda\, s_n^z \sin[\gamma\mathbf{S}_a^z] + \csc\lambda \sin\gamma\left(s_n^+\mathbf{S}_a^- + s_n^-\mathbf{S}_a^+\right) \tag{47}$$

that form the higher-spin transfer matrices of the XXZ model [8]. Here, $\mathbb{1}$ is the identity operator, $s_n^z$ and $s_n^\pm = s_n^x \pm is_n^y$ act on the $n$-th spin-1/2 degree of freedom, while operators in bold act over the auxiliary space, labeled by index a. They generate a complex spin-$s$ algebra, also known as quantum group $\mathcal{U}_q(sl_2)$, where $q = \exp(i\gamma)$. Focusing on root-of-unity $q$ (i.e., $q^k = 1$, for some $k \in \mathbb{N}$, which yields $\gamma = \ell\pi/k$, $\ell \in 2\mathbb{N}$), we choose a *semicyclic* representation

$$\begin{aligned} \mathbf{S}^z &= \sum_{k=0}^{m-1}(s-k)|k\rangle\langle k|, \\ \mathbf{S}^+ &= \sum_{k=0}^{m-2}\frac{\sin[\gamma(2s-k)]}{\sin\gamma}|k\rangle\langle k+1|, \\ \mathbf{S}^- &= \sum_{k=0}^{m-2}\frac{\sin[\gamma(k+1)]}{\sin\gamma}|k+1\rangle\langle k| + \beta|0\rangle\langle m-1| \end{aligned} \tag{48}$$

of the quantum group. Its dimension is set by the order of the root of unity, defined as $m = \min\{k \in \mathbb{N} \mid q^k = 1\}$. The oscillatory behaviour of the dynamical symmetries

$$Y(\lambda) = (\csc \gamma)^2 \partial_\beta \tilde{T}(\lambda)\big|_{\beta,s=0} \tag{49}$$

originates in the absence of the lowest-weight[4] state, i.e., $\mathbf{S}^-|m-1\rangle = \beta|0\rangle$. Before examining them in detail, we state several important properties:

1. They can be computed according to Eq. (49) only for odd orders $m$ of the root-of-unity parameter $q$ (see, however, Ref. [41] for the details on how to treat the anisotropies parametrized by the root-of-unity $q$ of even order).

2. They satisfy

$$\mathrm{U}^{-t} Y(\lambda) \mathrm{U}^t = e^{ihmt} Y(\lambda), \qquad t \in \mathbb{N}. \tag{50}$$

3. For $|\mathrm{Re}\lambda - \pi/2| < \pi/(2m)$ they are extensive (pseudo-local).

The first property is a consequence of the fact that the auxiliary representation (48) is only irreducible for odd $m$. A detailed discussion of this peculiar character of the semicyclic representations has been presented in Ref. [41]. For simplicity, we will restrict ourselves to the set of anisotropies, given in Eq. (30). Regarding the second property note that, in the absence of magnetic fields $Y(\lambda)$ are exactly conserved. This follows from the specific type of Yang-Baxter equation, the so-called RLL relation

$$R_{1,2}(\lambda - \mu)\mathbf{L}_{a,1}(\lambda)\mathbf{L}_{a,2}(\mu) = \mathbf{L}_{a,2}(\mu)\mathbf{L}_{a,1}(\lambda)R_{1,2}(\lambda - \mu), \tag{51}$$

which implies $[\tilde{T}(\lambda), T(\mu)] = 0$, provided that $h = 0$. We proceed to examine the second and the third property in detail.

## B.1  Explicit form of the dynamical symmetries

Due to cyclicity of the trace over the auxiliary space in Eq. (46), the derivative on $\beta$ can always be translated to the leftmost position in the string of Lax operators. Then, using

$$\partial_\beta \mathbf{L}_{a,n}(\lambda)|_{\beta,s=0} = \csc \lambda \sin \gamma \, s_n^+ |0\rangle_a \langle m-1|_a \tag{52}$$

and denoting $\mathbf{L}_{a,n}^0(\lambda) = \mathbf{L}_{a,n}(\lambda)|_{\beta,s=0}$, the operator (49) can be rewritten as

$$Y(\lambda) = \csc \gamma \sum_{n=0}^{N/2-1} \Big\{ \csc(\lambda_+) \hat{\mathcal{S}}^{2n} \big[ s_1^+ \langle m-1|_a \overbrace{\mathbf{L}_{a,2}^0(\lambda_-)\mathbf{L}_{a,3}^0(\lambda_+)\ldots\mathbf{L}_{a,N}^0(\lambda_-)}^{\text{string of Lax operators}} |0\rangle_a \big] +$$

$$+ \csc(\lambda_-) \hat{\mathcal{S}}^{2n+1} \big[ s_1^+ \langle m-1|_a \mathbf{L}_{a,2}^0(\lambda_+)\mathbf{L}_{a,3}^0(\lambda_-)\ldots\mathbf{L}_{a,N}^0(\lambda_+)|0\rangle_a \big] \Big\}. \tag{53}$$

Here and below, symbol $\hat{\mathcal{S}}$ denotes the conjugation by a one-site lattice shift, e.g., $\hat{\mathcal{S}}(s_n^\alpha) = s_{n+1}^\alpha$, for which the periodic boundary conditions imply $\hat{\mathcal{S}}^N = 1$.

We observe that, in each term, the highest-weight and the lowest-weight vectors in the auxiliary space have to be coupled by the *string* of Lax operators. For any $n \in \{1, 2, \ldots N\}$ we have (recall that $\beta = s = 0$)

$$\mathbf{L}_{a,n}^0(\lambda_\pm)|0\rangle_a = \mathbb{1}|0\rangle_a + \sin \gamma \csc(\lambda_\pm) s_n^+ |1\rangle_a, \tag{54}$$

---

[4]The state $|m-1\rangle$ is termed the lowest-weight state, if destroyed by the spin lowering operator, that is, when $\mathbf{S}^-|m-1\rangle = 0$.

so the string of Lax operators either lowers the auxiliary state and produces a spin raising operator $s_n^+$, or leaves the highest-weight state $|0\rangle_a$ in the auxiliary space intact, meanwhile contributing the identity operator $\mathbb{1}$ to the action on the total Hilbert space. This results in

$$Y(\lambda) = \sum_{n=0}^{N/2-1} \left\{ \sum_{r=\lfloor m/2 \rfloor}^{N/2-1} \left( \hat{\mathcal{S}}^{2n} \big[ q^{[2r+1,-]}(\lambda) \big] + \hat{\mathcal{S}}^{2n+1} \big[ q^{[2r+1,+]}(\lambda) \big] \right) + \right.$$
$$\left. + \sum_{r=\lceil m/2 \rceil}^{N/2} \left( \hat{\mathcal{S}}^{2n} \big[ q^{[2r,-]}(\lambda) \big] + \hat{\mathcal{S}}^{2n+1} \big[ q^{[2r,+]}(\lambda) \big] \right) \right\}, \tag{55}$$

where the local densities read

$$q^{[2r+1,\pm]}(\lambda) = \csc(\lambda_\mp)^2 s_1^+ \langle m-1|_a \mathbf{L}_{a,2}^0(\lambda_\pm) \mathbf{L}_{a,3}^0(\lambda_\mp) \dots \mathbf{L}_{a,2r}^0(\lambda_\pm) |1\rangle_a s_{2r+1}^+,$$
$$q^{[2r,\pm]}(\lambda) = \csc(\lambda_+) \csc(\lambda_-) s_1^+ \langle m-1|_a \mathbf{L}_{a,2}^0(\lambda_\pm) \mathbf{L}_{a,3}^0(\lambda_\mp) \dots \mathbf{L}_{a,2r-1}^0(\lambda_\mp) |1\rangle_a s_{2r}^+. \tag{56}$$

Except on the first $2r + 1$ (respectively $2r$) sites, these local densities act trivially, i.e., as identities. Importantly, they contain a surplus of $m$ spin raising operators $s_n^+$ acting on the physical degrees of freedom, as the string of Lax operators still needs to connect the states $|1\rangle$ and $|m-1\rangle$ in the auxiliary space. This can only be achieved via the auxiliary spin operator $\mathbf{S}^-$ [see the representation (48)], which is coupled to $s_n^+$ in the Lax operator (47). Local densities with a surplus of less than $m$ spin raising operators vanish, explaining the lower boundaries on the sums in Eq. (55). Due to the surplus of exactly $m$ spin raising operators in each term of $Y(\lambda)$ we now have

$$\mathrm{U}^{-t} Y(\lambda) \mathrm{U}^t = e^{iht \sum_{n=1}^N s_n^z} Y(\lambda) e^{-iht \sum_{n=1}^N s_n^z} = e^{ihmt} Y(\lambda), \tag{57}$$

where relation $[s_n^z, s_n^+] = s_n^+$ has been used. We proceed to examine the extensivity of dynamical symmetries $Y(\lambda)$.

## B.2 Extensivity of the dynamical symmetries

For simplicity, let us consider infinite-temperature susceptibilities, so that the state entering the inner product (2) corresponds to the featureless identity matrix $\rho = 2^{-N} \mathbb{1}$, while the inner product itself becomes of the Hilbert-Schmidt type. Being interested in the thermodynamic limit, we rescale it by the system size:

$$\langle A, B \rangle = \lim_{N \to \infty} \frac{1}{N} \left( \langle A^\dagger B \rangle - \overline{\langle A \rangle} \langle B \rangle \right). \tag{58}$$

To determine the extensivity of the dynamical symmetries, we need to consider the kernel of overlaps

$$\mathcal{Y}(\lambda, \mu) = \langle Y(\bar{\lambda}), Y(\mu) \rangle \tag{59}$$

that will be referred to as the Hilbert-Schmidt kernel. For ease of notation we have conjugated the spectral parameter in the first factor of the inner product.

The local densities (56) of $Y(\lambda)$ are orthogonal w.r.t. the Hilbert-Schmidt inner product; their overlaps vanish if the corresponding supports do not match perfectly. The Hilbert-Schmidt kernel is thus

$$\mathcal{Y}(\lambda, \mu) = \frac{1}{2} \sum_{r=1}^\infty \sum_{s \in \{+, -\}} \left[ \langle q^{[2r,s]}(\bar{\lambda}), q^{[2r,s]}(\mu) \rangle + \langle q^{[2r+1,s]}(\bar{\lambda}), q^{[2r+1,s]}(\mu) \rangle \right]. \tag{60}$$

Its computation is further facilitated by employing the matrix product structure of the local densities. In particular, we can define the auxiliary transfer matrix

$$\mathbb{T}_{a_1,a_2}(\lambda,\mu) = \frac{1}{2}\text{tr}_n\Big([\mathbf{L}^0_{a_1,n}(\lambda)]^T\mathbf{L}^0_{a_2,n}(\mu)\Big), \tag{61}$$

where $(\bullet)^T$ denotes the partial transposition of the operators over the physical degrees of freedom, namely those, indexed by $n$ in Eq. (47). Two indices $a_1$ and $a_2$ denote two copies of the auxiliary space.

The auxiliary transfer matrix now facilitates the computation of overlaps in the Hilbert-Schmidt kernel (60), for example

$$\langle q^{[2r,\pm]}(\bar{\lambda}), q^{[2r,\pm]}(\mu)\rangle = \frac{\langle m-1, m-1|\big[\mathbb{T}(\lambda_{\mp},\mu_{\pm})\mathbb{T}(\lambda_{\pm},\mu_{\mp})\big]^{r-1}|1,1\rangle}{4\sin(\lambda_+)\sin(\lambda_-)\sin(\mu_+)\sin(\mu_-)}, \tag{62}$$

while the extensivity of the dynamical symmetries $Y(\lambda)$ now depends on the spectrum of its projection onto the subspace $\mathcal{W} = \text{lin}\{|k,k\rangle \mid 1 \leq k \leq m-1\}$ that is invariant under its action, namely $\mathbb{T}(\lambda,\mu)\mathcal{W} \subset \mathcal{W}$. In particular, the sums over the support-size index $r$ in Eq. (60) converge, if the spectrum lies inside the unit circle in the complex plane. Linear extensivity of dynamical symmetries $Y(\lambda)$, i.e., finiteness of $\mathcal{Y}(\lambda,\mu)$, now follows from the following observation, proven in Ref. [38]:

**Proposition 1** *For $|\lambda - \pi/2| < \pi/(2m)$ the eigenvalues of the auxiliary transfer matrix $\mathbb{T}(\lambda,\mu)$ projected onto its invariant subspace $\mathcal{W} = \text{lin}\{|k,k\rangle \mid 1 \leq k \leq m-1\}$ are strictly below 1 in the absolute value.*

Evidently, the imaginary shifts of the spectral parameters by $\pm\delta/2$ do not cause violation of this condition as it constraints only the real part of $\lambda$. The explicit form of the Hilbert-Schmidt kernel $\mathcal{Y}(\lambda,\mu)$ can be conjectured, similarly as was done in Ref. [35], and is given in Eq. (33). It has been extensively numerically and analytically checked that it also reproduces the normalized inner product of the semicyclic symmetries in the continuous-time limit $\delta \to 0$; see Ref. [41].

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
