# Peer review of "Rigorous bounds on dynamical response functions and time-translation symmetry breaking"

_SciPost Physics, doi:SciPost Phys. 9, 003 (2020)_

## Round 1 · Referee Report · Anonymous · 2020-5-10

Strengths

1- the paper is clearly written;
2- the breaking of time translation symmetry is an extremely timely and interesting topic;
3- the mathematical aspects are rigorous although remain accessible to physicicst in the field;
4- the bounds are checked against numerics and are shown to be saturated;

Weaknesses

1- it is unclear if the discussion is fully general or it is focused on the behavior of specific integrable cases
2- the relevant point of the stability under perturbation is only mentioned in the conclusion

Report

This work considers the behavior of response functions, obtained by perturbing a system initially prepared at its thermal equilibrium (or in a stationary state). The approached followed retraces the construction of the celebrated Mazur's bound for dynamical correlation functions. Instead of considering conserved quantities of the system (i.e. operators commuting with the time-evolution generator), here the focus is on dynamical symmetries which generalise static ones: operators which are are eigenvectors for the adjoint action of the Hamiltonian. The authors are able to derive exact bounds which ensure that in the presence of dynamical symmetries and non-trivial overlap with relevant operators, one will observe persistent oscillations as measured by the absence of decay of the dynamical response functions.

Overall, the paper is dealing with a very timely topic which is the breaking of time-translation symmetry and the existence of time crystals. Additionally, the paper is well-written and contains rigorous results and a mathematical framework to study theoretically the occurrence of this phenomenon. I also appreciate the presence of a comparison with numerical simulations.
I recommend the publication on Scipost; I have a few questions and comments, whose answer could improve the readability and the clarity of the manuscript.

Requested changes

1- at the end of the introduction, the authors comment about the relation between time-translation symmetry breaking and the existence of extensive dynamical symmetries. Is it true in general that all forms of time-translation symmetry breakings do require the existence of dynamical symmetries as given in Eq. 12?
2-In Eq. (1), I could not understand if the $\rho$'s stand for the normalized density matrix or the unnormalized one $\rho(\beta) = e^{-\beta H}$. Please clarify.
3- Although this is well-known in the literature, can you please add some comments (or a precise reference) about why Eq. (2) is a well-defined scalar product ? In particular, since it is used a lot in the rest of the paper, why $\langle A, A\rangle \geq 0$?
4- Why in Eq. (5) you only consider the diagonal case where both $A$ and $B$ have the same $k$ component? Would it be relevant to treat the off-diagonal case?
5- In section 3, I think it would help a lot to have a couple of explicit examples of local / extensive dynamical symmetries, beyond their mathematical definitions.
6- An important point that I see is: what could be the physical origin of a relation like Eq. (9)? I understand that it can come out from explicit contruction of integrable models (as shown in Sec. 5), but beyond this is there a physical context which would lead to this "dynamical symmetries"? For conserved quantities, we know that they result from the Noether's theorem and a physical (or gauge) symmetry. What about dynamical symmetries discussed here? It would be important to know if they can come out in physical systems because of general principles.
7- Can you clarify what is the origin of the conjecture in Eq. 33?

  • validity: top
  • significance: high
  • originality: high
  • clarity: high
  • formatting: excellent
  • grammar: excellent

Author:  Marko Medenjak  on 2020-06-04  [id 845]

(in reply to Report 1 on 2020-05-10)
Category:
answer to question

We thank the referee for their positive report and in particular for the thorough and constructive evaluation, as well as questions and comments, which we address below.

"1- at the end of the introduction, the authors comment about the relation between time-translation symmetry breaking and the existence of extensive dynamical symmetries. Is it true in general that all forms of time-translation symmetry breakings do require the existence of dynamical symmetries as given in Eq. 12?"

There exists a set of systems with quantum scarred states where TTS breaking occurs only for special initial condition, which does not require the existence of dynamical symmetries. However, if TTS breaking occurs with respect to some stationary ensemble it implies the existence of dynamical symmetries that are extensive with respect to that ensemble.

"2- In Eq. (1), I could not understand if the ρ's stand for the normalized density matrix or the unnormalized one ρ(β)=exp(−βH). Please clarify."

ρ stands for the unnormalized density matrix, as introduced in the inline equation before Eq. (1). Instead the averages of observables are normalized, as described immediately after Eq. (1). We have now explicitly stated in the manuscript that this is an unnormalized state.

"3- Although this is well-known in the literature, can you please add some comments (or a precise reference) about why Eq. (2) is a well-defined scalar product ? In particular, since it is used a lot in the rest of the paper, why ⟨A,A⟩≥0?"

The reference has been added.

"4- Why in Eq. (5) you only consider the diagonal case where both A and B have the same k component? Would it be relevant to treat the off-diagonal case?"

We assumed that the Hamiltonian H is translationally invariant, implying that the off-diagonal elements are 0. We added the comment to the manuscript.

"5- In section 3, I think it would help a lot to have a couple of explicit examples of local / extensive dynamical symmetries, beyond their mathematical definitions."

We have added a simple example concerning the spin precession, as well as a more general comment on the dynamical symmetries that arise from a possible non-abelian structure of local conservation laws.

"6- An important point that I see is: what could be the physical origin of a relation like Eq. (9)? I understand that it can come out from explicit contruction of integrable models (as shown in Sec. 5), but beyond this is there a physical context which would lead to this "dynamical symmetries"? For conserved quantities, we know that they result from the Noether's theorem and a physical (or gauge) symmetry. What about dynamical symmetries discussed here? It would be important to know if they can come out in physical systems because of general principles."

They can arise as a consequence of non-abelian conservation laws that form a closed algebra. For instance, assume that the system possesses conserved quantities X and Y, [H,X]=0, [H,Y]=0, satisfying [X,Y]=Y. Then Y will be a dynamical symmetry for the Hamiltonian H'=H+X. The discussion has been added as another example in section 3.

"7- Can you clarify what is the origin of the conjecture in Eq. 33?"

The conjecture is guessed similarly as in Ref. [35], by considering the action of the auxiliary transfer matrix, given in Eq. (61) of the Appendix. Due to Proposition 1 (Appendix) the operatorial sum in Eq. (60) converges. One then needs to compute its matrix element using a fifth-order recurrence relation determined by the elements of the auxiliary transfer matrix (see Refs. [7], [41] and [35]). The solution can be guessed so as to reproduce the various limits in which solutions exist and can be proven (for example the continuous-time limit in Ref. [41]). Then, the result is compared to the numerical evaluation of the overlap. The word "numerically", as well as a short footnote have been added.

We would like to thank the referees again, for their examination of the manuscript.

---

## Round 1 · Referee Report · Anonymous · 2020-5-17

Strengths

This is an interesting contribution to the literature on dynamics of integrable systems. It builds on some recent ideas about persistent oscillations in integrable spin chains, and works out the details in a very clear way. I do not see any weaknesses, and think the paper should be published as is.

Report

I think this paper offers a very clear discussion of dynamical response, and especially its relation to dynamical symmetries in integrable spin chains (building on some previous work concerning "time crystals" and other persistent oscillations in these systems). I found the discussion very clear and have no significant comments.

  • validity: high
  • significance: high
  • originality: high
  • clarity: high
  • formatting: good
  • grammar: good

Author:  Marko Medenjak  on 2020-06-04  [id 846]

(in reply to Report 2 on 2020-05-17)
Category:
remark

We thank the referee for a very positive response.

---

## Editorial Decision

published